# Intelligent Grazing UAV Based on Airborne Depth Reasoning

Wei Luo [1,2,3,4], Ze Zhang [1], Ping Fu [5], Guosheng Wei [1], Dongliang Wang [2,*], Xuqing Li [1,3,4], Quanqin Shao [2,6], Yuejun He [1,3,4], Huijuan Wang [1], Zihui Zhao [1,3,4], Ke Liu [1,3,4], Yuyan Liu [1,3,4], Yongxiang Zhao [1], Suhua Zou [1] and Xueli Liu [1]

1   North China Institute of Aerospace Engineering, Langfang 065000, China
2   Key Laboratory of Land Surface Pattern and Simulation, Institute of Geographic Sciences and Natural Resources Research, Chinese Academy of Sciences, Beijing 100101, China
3   Aerospace Remote Sensing Information Processing and Application Collaborative Innovation Center of Hebei Province, Langfang 065000, China
4   National Joint Engineering Research Center of Space Remote Sensing Information Application Technology, Langfang 065000, China
5   Key Laboratory of Advanced Motion Control, Fujian Provincial Education Department, Minjiang University, Fuzhou 350108, China
6   University of Chinese Academy of Sciences, Beijing 101407, China
*   Correspondence: wangdongliang@igsnrr.ac.cn

**Abstract:** The existing precision grazing technology helps to improve the utilization rate of livestock to pasture, but it is still at the level of "collectivization" and cannot provide more accurate grazing management and control. (1) Background: In recent years, with the rapid development of agent-related technologies such as deep learning, visual navigation and tracking, more and more lightweight edge computing cell target detection algorithms have been proposed. (2) Methods: In this study, the improved YOLOv5 detector combined with the extended dataset realized the accurate identification and location of domestic cattle; with the help of the kernel correlation filter (KCF) automatic tracking framework, the long-term cyclic convolution network (LRCN) was used to analyze the texture characteristics of animal fur and effectively distinguish the individual cattle. (3) Results: The intelligent UAV equipped with an AGX Xavier high-performance computing unit ran the above algorithm through edge computing and effectively realized the individual identification and positioning of cattle during the actual flight. (4) Conclusion: The UAV platform based on airborne depth reasoning is expected to help the development of smart ecological animal husbandry and provide better precision services for herdsmen.

**Keywords:** precision grazing; intelligent UAV; cattle monitoring; YOLOv5; Inception V3; LSTM

## 1. Introduction

The application of precision grazing technology may promote the more dynamic management of grazing ruminants, from the macro level management to the individual level management of animals on the pasture. Studies in the past ten years on the wide application of GPS collars, Bluetooth ear tags, electronic fences and other devices have proven that the sensor and information technology development assisted in enhancing the monitoring of grazing animals, especially cattle [1–4]. Demands from consumers as well as from exporters require that cattle shall be identified and traceable [5], and many countries have developed legal mandatory frameworks [6] which revolve around the national databases and ear tagging [7–9]. Bovine identification heavily depends on such a tagging approach, which is not effective in many cases compared with branding, tattooing [5] or electronic solutions [10]. The major reason is that ear tags can easily be lost and lead to physical injury [11]. In addition, there are animal welfare concerns in terms of the ear tagging [12,13].To that end, coat-pattern-based visual bovine identification exhibits

an automated and non-intrusive nature, which assisted in improving the farm efficiency as well as promoting animal welfare.

The rapid development of UAV (Unmanned Aerial Vehicle) technology provided a new and low-cost tool for animal investigation. Compared with the traditional methods (such as ground counting and man–machine survey), it has more advantages, such as relative low risk and low cost [14,15], though it is still in the exploration stage. Since it was reported that it was used in the investigation of American alligators and waterfowl in 2006 [16], UAVs made some progress in animal identification, tracking [17,18], size measurement [19] and behavior investigation [18,20]. Due to the low cost of UAV images and the fact that the resolution can be adjusted according to the altitude (up to 2 mm), UAV images can be used to identify large animals such as African elephants, giraffes [18], manatees [17], cattle, and sheep [21], as well as animals as small as penguins, albatross cubs [22], Canadian geese [23], and even flying insects such as bumblebees [24].

Deep learning is a subfield of machine learning that uses neural networks to automate feature extraction, permitting raw data to be input into a computer and creating high-level abstractions to inform decisions in classification, object detection, or other problems [25]. The majority of recent advances in computer vision and object detection have been made with convolutional neural networks (CNNs) [26,27]. CNNs ingest data in multidimensional arrays (e.g., 1D: text sequences; 2D: imagery or audio; 3D: video) and scan these arrays with a series of windows that transform the raw data into higher level features that represent the original input data through multiple layers of increasing abstraction. CNN applications within ecology are becoming widespread, including the rapid development of species identification tools [28]. For example, Norouzzadeha et al. [29] were able to identify 48 different animal species from camera traps in the 3.2 million image Snapshot Serengeti dataset with 93.8% accuracy, similar to the accuracy of crowdsourced identifications, saving nearly 8.4 years of human labeling effort. More recently, Gray et al. [30] used a CNN to detect and enumerate olive ridley turtles in the nearshore waters of Ostional, Costa Rica, identifying 8% more turtles in imagery than manual methods with a 66-fold reduction in analyst time.

Animal biometric recognition technology was adopted with the advantages of the variability and uniqueness of fur patterns, phonation, movement dynamics and body shape, and defined the animal categories of interest in a highly objective, comparable and repeatable way by calculating and interpreting the information about animal appearance [31]. The unique patterns and special markers have been used for computer-aided individual recognition, including the spot patterns in manta rays [32], penguins [33] and whale sharks [34] and the stripe patterns in tigers [35]. The livestock biometrics research includes cattle [36], sheep [37,38], horses [39] and pigs [40]. To obtain the latest performance, the individual ID component is based on the latest CNN-grounded biometric work [38]; thereinto, a Long-Term Recurrent Convolutional Network (LRCN) assists in analyzing the detected temporal stacks regarding the region of interest (ROI). Finally, the temporal information is integrated and mapped to the information vector of individual animals through a Long Short-Term Memory (LSTM) unit.

It is difficult for the current processing board of airborne images to achieve large target solving tasks due to its limited computing ability. YOLOv5 is a type of target recognition network with very light weight, is capable of dealing with the low efficiency exhibited by a full convolution model network and can ensure the effect of classification. Many public datasets have been verified to confirm its accuracy, which is the same as that of the Efficient Det and the YOLOv4, but its model size only takes up 1/10 of the latter two approaches [41]. YOLOv5 with edge computing shall be ideally conducted on UAVs and unmanned ships, as well as other platforms [42]. Such architecture achieved the light-weight onboard operation on the one hand and ensured higher efficiency networks that exhibit larger computational room on the other hand.

Therefore, the goal of the present work was to demonstrate the following: we pre-train and improve the YOLOv5 detection model in combination with the expanded dataset to

promote the recognition accuracy of cattle. The UAV tracks the cattle according to the prediction frame through the KCF tracking algorithm. Finally, the predicted cattle are distinguished by the LRCN recognition frame. All the above algorithms can be realized by the edge computing unit on the UAV combined with the flight control module and communication architecture.

## 2. Materials and Methods

### 2.1. Introduction to Studied Area and Study Object

The studied area is Qilian County, which belongs to the Haibei Tibetan Autonomous Prefecture of Qinghai Province, and is located in the hinterland of the middle Qilian Mountains, with Hexi Corridor in the north and a lake-circumnavigation passage in the south (see Figure 1a). It is adjacent to the Qilian Mountain grassland, one of the six major grasslands in China. The average altitude of the territory is 3169 m, the average annual temperature is 1 °C and the annual precipitation is about 420 mm. It belongs to the typical plateau continental climate. Because it is a unique geographical location and ecological environment, the area is very rich in animal and plant resources, especially developed animal husbandry and is a large animal husbandry county. This place was selected for AI-based precision grazing technology research as it has great significance for the protection of large herbivores and restoration of ecological vulnerability in the combined erosion area of the Qinghai Tibet Plateau [43].

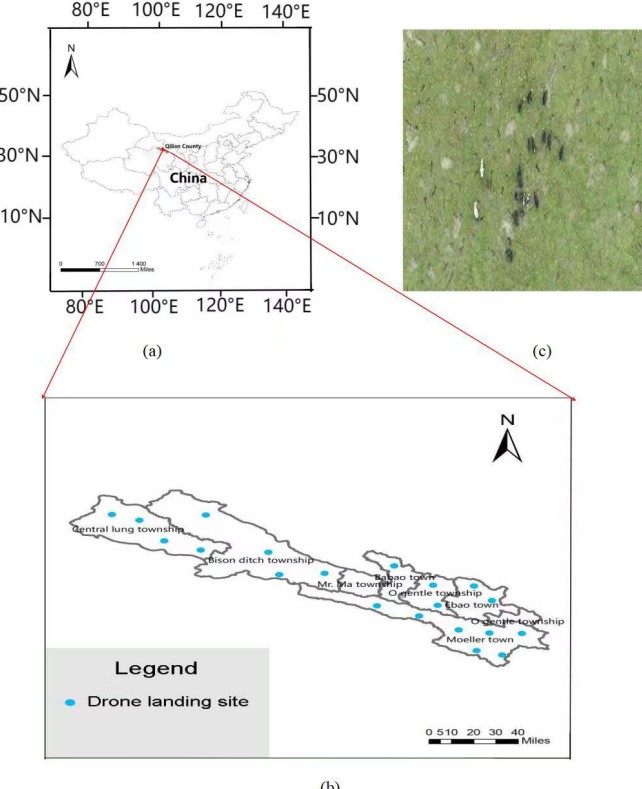

**Figure 1.** The studied area selected in this paper: (**a**) location of Qilian County; (**b**) distribution of UAV landing points in Qilian County; (**c**) aerial image of study area.

In August, the author and members of the research group went to Qilian County to carry out an aerial survey by UAV. A total of 20 sorties were flown in three days (see Figure 1b), with about half an hour every flight and a flight height of about 100 m. The total area covered by aerial photography reached 2100 km$^2$ and we collected a large number of video data and forward remote sensing images in the studied area (see Figure 1c). In this paper, we selected domestic cattle as research objects. Based on the seven elements

of remote sensing interpretation, a tag library for feature extraction was constructed [44], which is summarized in Table 1.

**Table 1.** Identification database of domestic cattle.

| Feature | Illustration |
|---|---|
| Tonal | With black, gray-black and other dark colors |
| Color | The main colors are white, black-white and black |
| Texture | A solid color or a plurality of solid color splicing |
| Size | The adult domestic cattle have a body length of about 1.6~2.2 m. For example, if the resolution is resolution, the individual length is more than 40~50 pixels. |
| Shape | The overall shape is nearly elliptic, rectangular. The ratio of length and width is mostly between 1.4:1 and 3:1. |
| Group image | 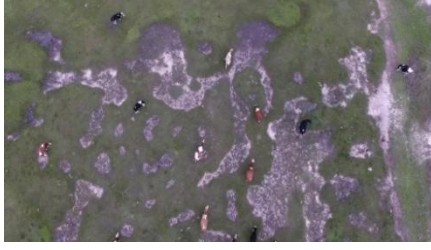 |
| Individual recognition paradigm | 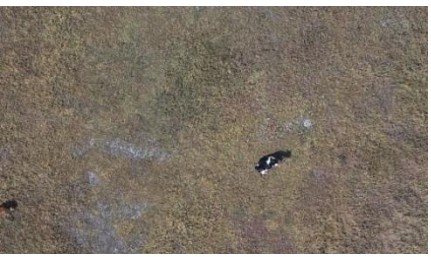 |
| Shape features | 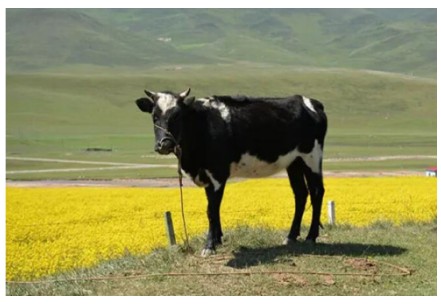 |

### 2.2. Study Workflow

In this study, this paper proposes a method of automatic identification of animal individuals based on deep learning to achieve the purpose of accurate animal husbandry. Real-time identification tracking of individuals was achieved by the improved YOLOv5 algorithm to generate corresponding ROI annotation frames or candidate boxes, the KCF target tracking algorithm for the trajectory recording and the LRCN prediction of the input sequence to generate the final prediction vector (see Figure 2).

We proposed a research framework for the collaborative design of software and hardware, which was integrated with the flexibility of software and the efficiency of hardware to achieve the detection and identification of animals. Through the deep learning algorithm mounted on the drone, we realized the animal detection (green), trajectory recording (blue) and individual prediction (red). The specific steps are as follows:

1. Data acquisition. The video streaming image data by controlling the P600 intelligent UAV equipped with a three-axis photoelectric pod to fly to a specified location are captured.

2. Animal detection and localization. The acquired video stream data were reshaped into $299 \times 299$ images, and processed by the improved YOLOv5 animal detection model, predicting the presence or absence of animals and mapping the detected animals to finally obtain an initialized ROI prediction frame.

3. Trajectory recording. It is possible to select the KCF kernel correlation filtering algorithm to track the object of interest when it is detected/recognized because of its advantages of high precision and high processing speed, both in terms of tracking effect and tracking speed. Through the KCF target tracking algorithm, the detection frame is determined to see whether the target animal is monitored. If it is detected, it is learned and tracked, or otherwise, the new frame is re-examined to find the animal of interest. The fast extraction of detected trajectories is helpful to obtain more accurate ROI annotation frames to be helpful to further extract more realistic visual features.

4. Generate space–time trajectories. The individual ROI annotation frames obtained from KCF were converted into a set of space–time trajectories.

5. Individual prediction. Both the weights of CNNs and Long Short-Term Memory model (LSTM) were shared across time, allowing real-time identification tracking of targets in the video. Each set of the space–time trajectories was rescaled as well as passed to an Inception V3 network until reaching layer 3 of the pool, where the visual features were extracted from the input frames and fed into an LSTM recurrent neural network, followed by being recombined with image frames as input to subsequent iteration frames based on a time-lapse data sequence. After processing this set of spatiotemporal trajectories, the whole input sequence can obtain ID final predictions via a layer with full connection.

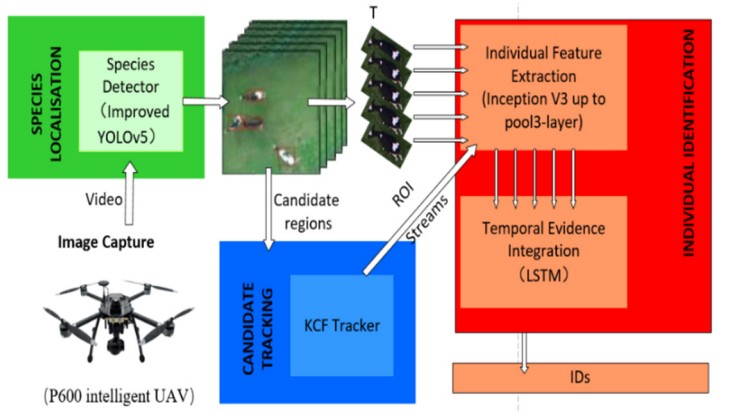

**Figure 2.** Proposed component pipelines.

### 2.3. Data Acquisition

The video acquisition in the studied area is realized through the P600 intelligent UAV (Figure 3a) produced by Chengdu Bobei Technology Co., Ltd., Chengdu, China and the photoelectric pod (Figure 3b upper) carried by P600. Prometheus 600 (P600 for short) is a medium-sized UAV development platform with the characteristics of large load, long endurance and scalability. It can be equipped with laser radar, onboard computer, three-axis photoelectric pod, RTK and other intelligent equipment to realize pod frame selection and tracking, laser radar obstacle avoidance and UAV position and speed guidance flight. P600 was equipped with a Q10F $10\times$ zoom single light pod with USB interface for P600 and developed its special Robot Operating System (ROS) driver, which can obtain real-time images of the pod in an airborne computer.

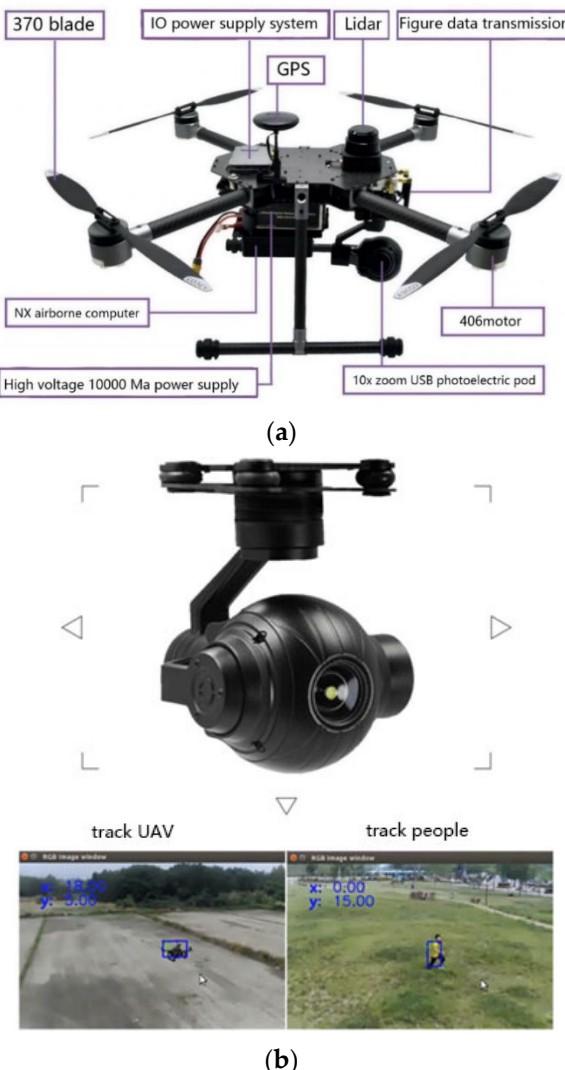

**Figure 3.** Data acquisition equipment for the study. (**a**) P600 intelligent UAV; (**b**) P600 equipped with intelligent photoelectric pod.

Furthermore, P600 can recognize, track and follow specific targets (human/vehicle/other UAVs, etc.) based on image vision through the built-in KCF frame tracking algorithm in the airborne system (Figure 3b bottom). It can even calculate the approximate distance between the robot and the tracking target by changing the size of the visible target frame. In addition to following the target, P600 can also adjust its position when the target approaches to always maintain a fixed distance from the target. In the process of intelligent pod tracking of the target, both the pod and UAV can achieve full autonomous control through ROS.

### 2.4. Hardware Communication Architecture

The overall communication framework of P600 is shown in Figure 4. It was adopted with the design of full body internal wiring + built-in flight control, leaving developers with a total of three layers of expansion space. Combined with the flight control expansion interface at the top layer of the UAV fuselage and the onboard computer at the bottom layer, sensors suitable for Px4 flight control or ROS can be freely added.

The onboard computer NX and Codev flight control communicate through serial port connection. The former sends any desired commands to the flight control based on Mavros, including desired position, desired speed and desired attitude. The onboard computer NX can obtain the image of the photoelectric pod through the USB port, run the KCF detection algorithm to detect and track the object and calculate the corresponding pod

control commands and UAV control commands. The pod control command is sent to the photoelectric pod through the serial port connected with the NX to make the pod lens rotate with the moving object. UAV control command is sent to Codev flight control via serial port through Mavros to control the UAV and then track the moving objects. The ground station computer can remotely access and view the desktop image of the NX end of the onboard computer through the Homer image data transmission and can also view the image of the photoelectric pod camera read by the NX of the onboard computer.

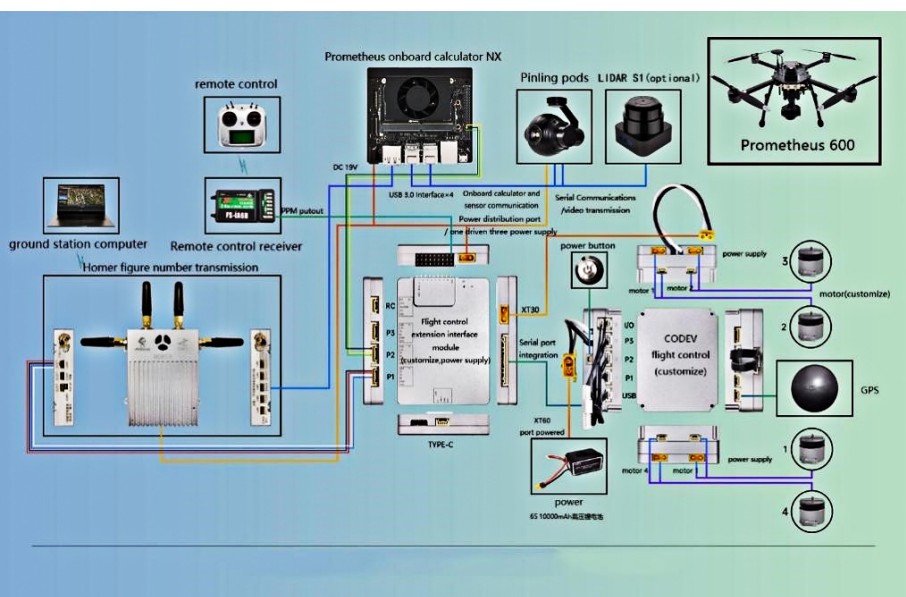

**Figure 4.** Hardware communication architecture for this study.

*2.5. Experimental Platform*

The onboard image processing board used in the experiment is the NVIDIA Jetson AGX Xavier embedded unmanned Intelligent Field platform launched by NVIDIA (Figure 5). This is a modular AI supercomputer which has a GPU for NVIDIA Volta with 512 CUDA cores. The CPU is provided with 8-core ARM V8.2, with strong AI computing power. It is powerful and compact in shape. Its performance is 20 times higher than the previous generation of NVIDIA Jetson TX2 platform (GUP is NVIDIA Pascal's 256-core CUDA-core architecture with quad-core ARM CPU). Its power consumption ratio is 10 times higher. NVIDIA JetPack and DeepStream SDK, as well as CUDA, CuDNN and TensorRT software libraries, can all be used to coordinate the creation and deployment of end-to-end AI robot applications. It is the most suitable for smart edge devices (robots, drones, smart cameras, etc.). The YOLOv5 model was first trained on a virtual machine, and then imported into the AGX Xavier processing board by using an SD card. Training the corresponding path of the model helps to obtain the processing result.

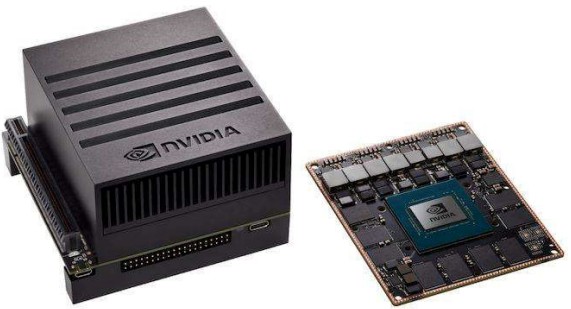

**Figure 5.** NVIDIA Jetson AGX Xavier embedded platform.

## 3. Airborne Depth Reasoning Network

### 3.1. Annotation and Augmentation of Training Data

A total of 10,000 remote sensing images of cattle taken by UAV were selected from the UAV survey area as the sample dataset, with a resolution of 1 cm and a size of 2048 × 1080 pixels. Among them, 7000 images were used as training sets, 2000 as test sets and 1000 as verification sets to verify the recognition results. In this study, 7000 images of UAV were labeled with professional labeling software *labelImg*, including boundary box labeling and cattle individual/category labeling. Boundary box annotation was performed to annotate boundary boxes in 7000 images, and the generated data were stored in XML format. The format of this dataset is VOC data format. Cattle individual/category marking: After the boundary box marking, the ROI area around the cattle was annotated and the individual cattle were required to be included in the boundary box during the target identification process. The XML document was used to record the coordinates of the upper left as well as lower right corners of the rectangular box.

In order to improve the stability of the training model, we extended this dataset. From July to August 2019, a total of 12,355 images of cattle were captured in Urumqi, Hami and Hulunbuir cities in Xinjiang. From June to September 2020, 12,701 cattle images and 37 cattle videos were obtained in Hulunbuir City, Inner Mongolia. In order to balance the individual balance of the total cattle in the image, we obtained 5501 Zhangjiakou and 5510 wild yaks by image synthesis, image cutting and image flipping on the original dataset. A total of 11,011 images of individual cattle were used as instance objects [45]. In addition, data enhancement, i.e., random shearing, rotation, scaling and flipping, was performed on the cattle boundary region dataset to generate multiple similar images, because, during the data collection process, the cattle can graze on the grassland at will. As a result, the time that is spent in the static acquisition system view exhibits individual differences, and also there is an imbalance in the number of individual cattle in the image among the total herd. For balancing the image number in training, the dataset was expanded by image synthesis. We selected the number of target instances as the largest number of original (non-synthetic) instances for any particular individual. Other images were synthesized by rotating the original image around the image center (x, y) by some random angles while maintaining the original image resolution to maintain the consistency of the dataset. In this study, data enhancement and data expansion were used to effectively reduce over-fitting, enhance the model stability and the generalization effectiveness and improve the identification effect of cattle individuals. Finally, UAV images were converted into a dataset in visual object format for the pre-training of the deep learning model.

### 3.2. Species Detection and Localization Based on Improved YOLOv5

YOLO (you only look once) acts as a single-stage object detection algorithm that detects faster than two-stage algorithms. YOLOv5 is the latest network architecture of the current YOLO series iteration, which was modified on the basis of YOLOv4 to enhance feature fusion capabilities and multi-target feature extraction capabilities, improve detection accuracy and to meet the needs of real-time image detection, and has been widely used in many fields. The improved YOLOv5 model was divided into three parts: backbone network, neck and output composition.

Backbone: Backbone networks are used for feature extraction. It replaced the first two layers with two RepVGG modules with the addition of improved CBAM modules to enhance feature extraction. The BottleneckCSP module maps the features of the base layer to two parts. The SPP module converts a feature map of any size into a fixed-size feature vector.

Neck: The neck network is primarily responsible for feature enhancement. The use of GSConv instead of the original Conv and the replacement of the BottleneckCSP module with the VoVGSCSP are used to preserve as many hidden connections as possible for improved model accuracy.

Output: After the neck network is mapped by fusing features, the output is responsible for predicting the features of the image, generating bounding boxes and predicting categories.

To improve the accuracy of species detection and localization, the original YOLOv5 model was improved. Figure 6 explains the improved model.

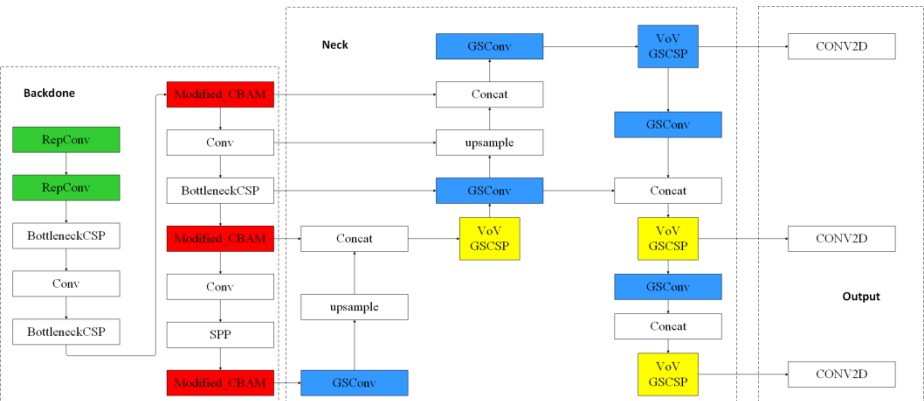

**Figure 6.** Improved YOLOv5 network structure. The green section indicates that backbone is optimized using RepVGG, the red part indicates the introduced improved attention mechanism model and blue and yellow indicate improvements to the neck layer of the slim-neck model.

1.   Modification of the anchor box size

The anchor box can be used to obtain a more accurate target bounding box by sampling many areas in the input image, followed by adjusting the area containing our target of interest, effectively limiting the range of predicted objects during training and accelerating the convergence of the model [46]. To obtain more accurate target information, the closer the aspect ratio of the anchor frame to the aspect ratio of the real bounding box, the better. However, due to differences in the size of individual animals in the drone image, the anchor frame size obtained by YOLOv5's original clustering cannot effectively cover the size of all animals, so the data need to be reclassified. The K-Means clustering algorithm can divide the dataset into several classes through intrinsic relationships. The same class exhibits a high similarity and different classes exhibit a low similarity, the corresponding center point of each sample data is given and the loss function corresponding to the clustering result is minimized by iteration. The study integrates the K-Means clustering algorithm that generates anchor box scales into the YOLOv5 algorithm.

2.   Improvements of neck layer

YOLOv5's neck layer is improved by slim-neck's model, as shown in Figure 6. In order to alleviate the current problem of high computational cost, the neck layer of YOLOv5 is improved by the slim-neck model proposed by Li et al. [47], which is capable of reducing the complexity of the model, while maintaining the recognition accuracy. The slim-neck architecture is divided into three models, GSConv, GSbottleneck and VoV-GSCSP. The GSConv model is adopted with deep-wise separable convolution (DSC) combined with standard convolution (SC), so that it can reduce the computational complexity through DSC and alleviate the problem of low recognition accuracy caused by low feature extraction and fusion capabilities of DSC through the SC model. As shown in Figure 7a, the information of the SC is generated through DWConv to perform the DSC operation, and the generated information is fused with the previous one. The VoV-GSCSP model is designed by a one-time aggregation method to improve the inference speed of the network model and maintain recognition capabilities, as shown in Figure 7b.

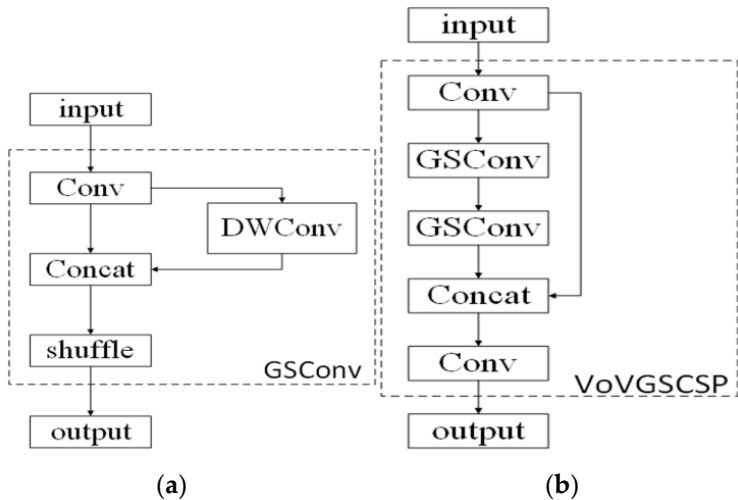

(a)             (b)

**Figure 7.** Slim-neck model. (**a**) GSConv model. (**b**) VoVGSCSP model.

3.  Introduction of attention mechanisms

For improving the model detection effect, the attention mechanism is introduced to enhance the feature representation of the CNN, so as to be focused on the key information of the task target in a large amount of information and reduce the attention to irrelevant information. Common attention models are the SE model, ECA model, CBAM model and so on. CBAM is a lightweight convolutional attention model that improves model performance at a fraction of the cost while being easily integrated into the existing network structures [42]. The CBAM model is combined with the two submodels of CAM and SAM, which can generate an attention feature map information in both the channel and space dimensions, and then multiply it with the previous feature map information to adaptively adjust the features and generate a more accurate feature map. In order to solve the situation, CBAM uses MLP structure to extract channel information and lose target information [48]. ECA-Net is used to replace CBAM's channel attention model. The improved CBAM model is shown in Figure 8.

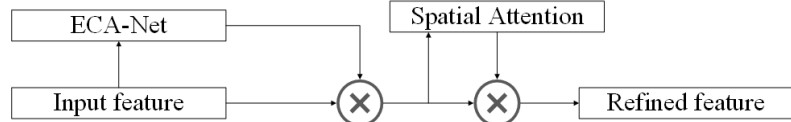

**Figure 8.** Improved CBAM model.

4.  Optimization of the backbone layer

RepVGG can be understood as a re-parameterized VGG. Using structure re-parameterization to obtain a reconstructed simple form of the inference model, the parameter reconstruction before and after its calculation results are mathematically consistent, so there is no loss of precision [49]. By filling and fusing into a $3 \times 3$ convolutional structure, the computing power of the hardware was fully utilized, so that the model inference speed was accelerated. To improve model performance, the backbone of YOLOv5 was optimized, and we changed the first two layers of the network to the RepConv layer to extract low-level semantic features, as shown in Figure 6.

### 3.3. Aerial Real-Time Photography LRCN Identification of Cattle

Different from the single still image of an environment, an event or a scene, video intrinsically offers another information dimension (i.e., temporal dimension) with regards individual identification. It is suggested that information from later frames should be incorporated into identification estimation, thus complementary information that is revealed

gradually can play its role [50]. In most cases, it is possible to track individual cattle in the herd videos well via the standard KCF tracking algorithm [51] when the above localization component species generate a good initialization ROI. Thus, if a cow appears in frame $f_i$, there is a large possibility for it to appear in $f_{i+1}$ (the frame rate is fully frequent in source footage). Considering these factors, the position and rotation of source footage captured by UAV can change, because winds, GPS inaccuracy, etc., can result in the change in viewpoint, object configuration and/or scale, while what is important is that it usually explains there are some more prominent visual features that can assist in the identification. Continually assessing the identity of an object with time when the parameters are changing supports the predictions of class, so that they are refined and improved iteratively.

Basically, LSTM networks are running on time-based data series, thus they run towards such task goals intrinsically. When evaluating the video and the image sequence of length *n*, it is required to consider the individual image frames sequentially. Specific to certain frame $f_i$, we considered the output from LSTM layer(s) as the input to layer(s) in the following iteration regarding frame $f_{i+1}$. As for the task case in the study, after processing the frame fn, a layer with full connection was adopted to generate the final class-prediction vector for the whole input sequence. An Inception V3 CNN was employed to input the extracted representations of a convolutional visual feature of input individual frames into an LSTM layer [52]. The approach Long-Term Recurrent Convolutional Networks (LRCNs) that combines CNNs and LSTMs was first developed by J. Donahue et al. [53] and is applied in the study to deal with the spatiotemporal identification. Figure 9 displays the standard LRCN pipeline.

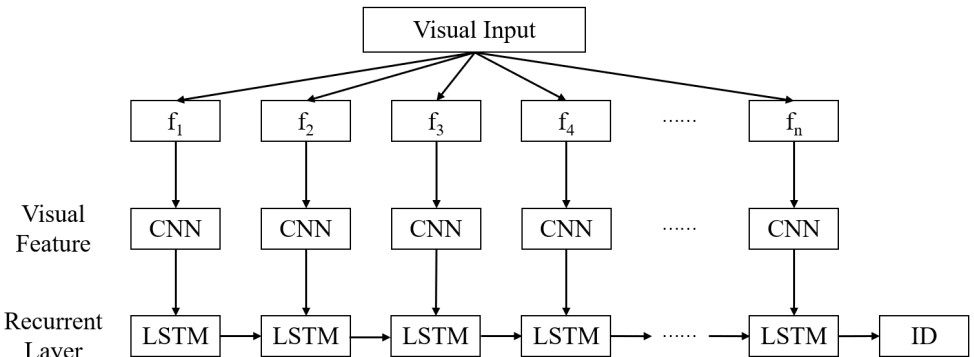

**Figure 9.** Recurrent convolutional architecture.

An unrolled identification refinement pipeline regarding an input video is on the basis of the LRCN architecture [48]. A CNN was employed to assist in extracting the visual features regarding input video frames $\{f_1, f_2, \ldots, f_n\}$ for input into an LSTM layer, which finally resulted in a ID prediction. The study states that such a core identification pipeline is capable of easily being integrated into an intact video processing architecture (Figure 2).

## 4. Results

### 4.1. Detection and Location of Cattle

In this study, we adopt the Pascal VOC matrix [29] by Everingham et al. as an evaluation protocol for verifying the false positives (FPs), the true positives (TP) and the false negatives (FNs). With one prediction bounding box corresponding to a single real bounding box, it is allowed to count the bounding box as TP if it has a maximum Intersection Over Union (IOU) and a certain solid bounding box and achieves the IOU threshold (0.8). In other cases, we treat the predicted bounding box as an FP. It is also allowed to treat the bounding box as FN when the IOU threshold (0.8) is achieved, and a combination of the actual bounding box and the predicted bounding box cannot be

achieved. The recall (R) and accuracy (P) are taken into account for evaluating the cattle prediction, defined as follows:

$$Precision = TP/(TP + FP) \tag{1}$$

$$Recall = TP/(TP + FN) \tag{2}$$

Recalls can help to more deeply learn cattle predicted coverage; however, the accuracy can be used to assess the accuracy of the total amount of projections. Since recall and accuracy only partially reflect model performance, the results are comprehensively evaluated by using the average accuracy (AP) and F1 scores, as defined as follows:

$$AP = \sum_{i=1}^{n} Precision_i(Recall_i - Recall_{i-1}), withRecall_{i=0} = 0 \tag{3}$$

$$F1 = (2 * R * P)/(R + P) \tag{4}$$

The algorithm's score threshold can be set to 0.8 for suppressing low score predictions. High score predictions are compared to surface facts for producing TP, FP, FN, accuracy, recall and AP.

A remote sensing image dataset of cattle was trained on GTX 1080 by using deep learning models (Fast-RCNN, YOLOv5, and improved YOLOv5), respectively. A total of 100 iterations of the data were performed, 100 models were annotated and the YOLOv5 model precision–recall curve was modified (Figure 10).

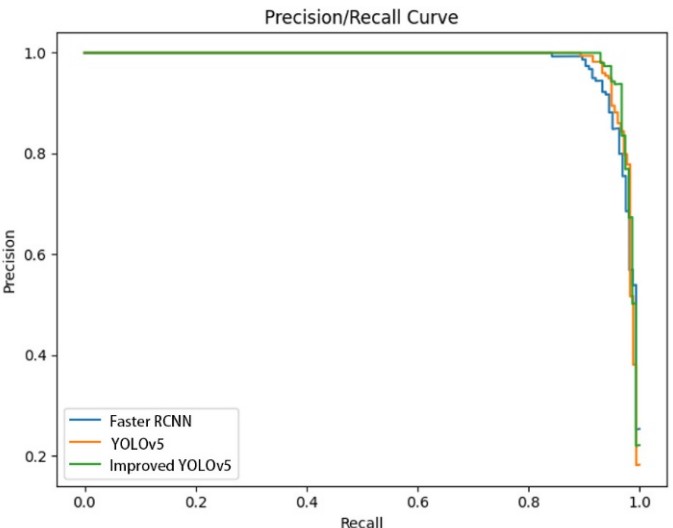

**Figure 10.** Improved YOLOv5 accuracy evaluation.

Compared with the Faster RCNN and the original YOLOv5, the improved YOLOv5 has both high accuracy and high recall when weighing accuracy and recall, indicating that the improved YOLv5 has better detection effect and better performance.

Based on the training results, the cattle were detected, as shown in Figure 11.

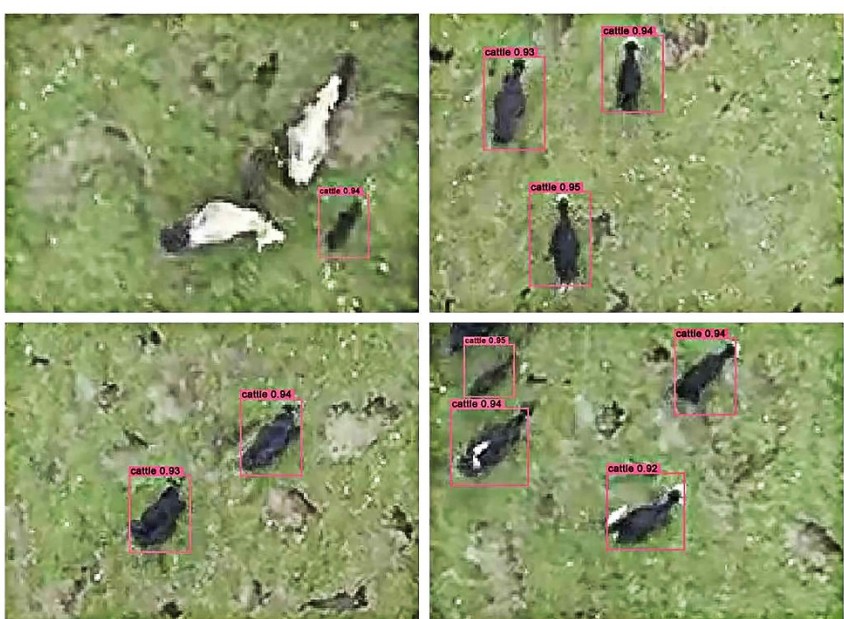

**Figure 11.** Cattle identification results.

*4.2. Accuracy Comparison*

We migrated our algorithm with the mainstream algorithms Faster RCNN and YOLOv5 to the development board for experimental comparison, use precision, recall rate and average accuracy to quantitatively analyze the experimental results generated by these models for further analyzing the proposed algorithm identification performance on the target of cattle. The specific data results are shown in Table 2. The improved YOLOv5 model has significant advantages over Faster RCNN in FPS and size and has certain advantages over YOLOv5 in precision and recall, which indicates the best overall performance. Therefore, it can effectively meet the task requirements of real-time detection and positioning of cattle.

**Table 2.** Dataset detection results.

| Network | FPS | Precision | Recall | Average Precision | Size of Model |
|---------|-----|-----------|--------|-------------------|---------------|
| Faster RCNN | 10.24 | 0.964 | 0.893 | 0.971 | 345 MB |
| YOLOv5 | 46.37 | 0.969 | 0.902 | 0.975 | 14.5 MB |
| Modified YOLOv5 | 43.63 | 0.984 | 0.921 | 0.983 | 15.2 MB |

*4.3. Video-Based LRCN Identification*

The dataset in the task is composed of ROI returned by YOLOv5 in the previous part, with 52,800 cropped image frames and 32 cattle with a total of 158 video instances. The video instance was divided into 40-frame-long spatiotemporal streams. When dividing the original dataset according to ROI, it generated 1320 tag streams, and each stream contains separate data. Then, these data were segmented according to the ratio of 9:1, used for data training and testing.

Inception V3 obtained from the ImageNet dataset was selected as the initialization network for recognition refinement. On this basis, the network was fine-tuned by using frames of 32 classes and 1188 training streams as input. In this case, after the third pooling layer through the network, a 2048 d unit vector was obtained as the output feature of the input image. Then, the representation of the convolutional frames served for training a single LSTM layer that had 512 units. The variation in training and test set accuracies over the course of 800 training epochs is shown in Figure 12.

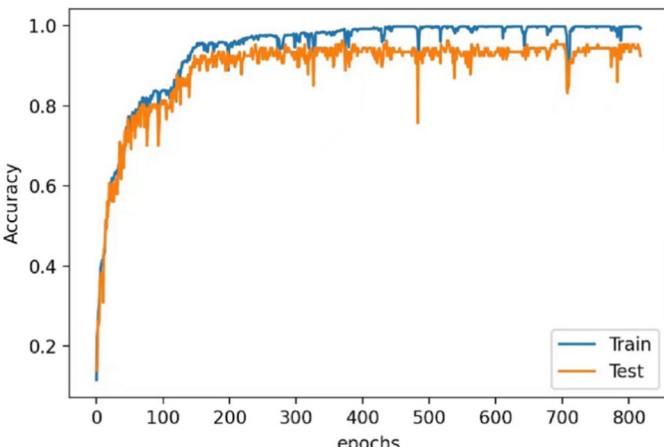

**Figure 12.** Individual recognition training: LSTM consisted of 512 units, recognition prediction accuracy at different stages. The training and test sets consisted of 1188 and 132 image streams.

In addition, for each prediction, an ordered vector [0, 1] of size |classes| = 32 was generated by using class confidence 2. The predicted class label denotes the index regarding the largest value in this vector. If the prediction matches with the true class label, the prediction is considered a positive sample. The precise recall curve of the recognition task is as shown in Figure 13.

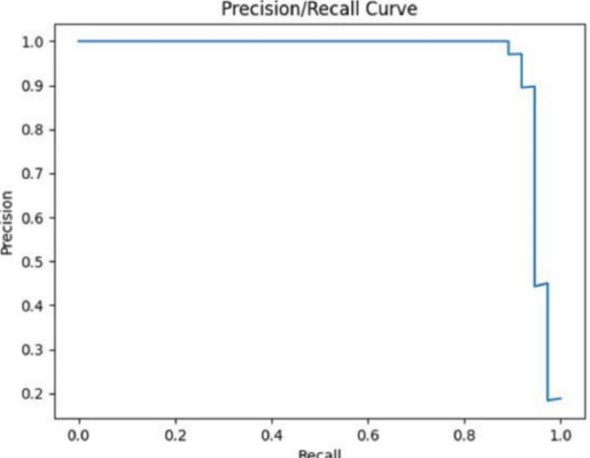

**Figure 13.** The precision-recall curves for the recognition task.

## 5. Discussion

The YOLOv5 network is composed of four architectures with different sizes (namely YOLOv5s, YOLOv5m, YOLOv5l and YOLOv5x). The size of the architecture varies with the depth and width of the network. In this study, considering the need to deploy the target detection model in a UAV and carry out real flight verification, YOLOv5s with the least parameters, the fastest speed and the lightest volume has become the first choice. However, it is inevitable to sacrifice accuracy while computing speed is fast. Therefore, we propose an improved YOLOv5 model: a K-means clustering algorithm serves for regrouping the anchor box for individual detection. The improved CBAM attention model was introduced to improve the attention to the target and reduce the influence of irrelevant information. By improving the partial optimization of the neck layer and backbone, the detection speed and accuracy can be further improved.

Compared with the Faster RCNN model, the improved YOLOv5 can reduce the parameters and complexity of the model. What is more, the FPS is improved by three times and the size is less than 5% of the latter. Compared with the original YOLOv5,

the precision of the model is promoted by 1.5% and the recall rate is improved by 1.9%. Although the progress is limited, the slight improvement in the identification rate of species categories is still helpful to ameliorate the effect of individually distinguished cattle under the setting that the number of cattle is usually huge. In addition, in the process of UAV data acquisition, image distortion, weather, exposure, resolution, terrain and other factors often affect the recognition effect. Therefore, we used data collected from different environments to expand the training dataset to further improve the generalization ability of the model.

According to the experimental data, the improved YOLOv5 has higher accuracy, recall rate and AP value. The overall performance is the best, which is very suitable for the real-time monitoring scene of cattle. The high-speed target detection model cannot only solve the image quickly, but also locate the cattle in time in the complex background. Although the Faster RCNN also has high accuracy, due to the characteristics of second-order network, its speed is far lower than that of first-order network, which cannot meet the requirements of real-time detection, and its deep network structure is not conducive to hardware deployment. YOLOv5 can be used to locate and identify targets based on the idea of regression. Thanks to its lightweight characteristics, it can solve the problem of hardware deployment as well as the problem of speed.

After preprocessing (such as cutting) the image frame after object detection, the LRCN network was introduced to recognize cattle individuals. LRCN is a network structure combined with CNN network and LSTM and has the ability to process single-frame pictures, image stream input and single-value prediction and time-series prediction output. Among them, the CNN part was adopted with the Inception V3 network, which uses a large number of parallel and dimensionality reduction structures to reduce the impact of structural changes on nearby models. The fusion of multi-scale feature spaces can avoid the loss of edge features and premature network fitting. Meanwhile, the relatively lightweight network structure can be more easily applied to mobile terminals.

## 6. Conclusions

This paper presents the improved YOLOv5 model for identifying cattle in the Qinghai Tibet Plateau, which has the following advantages: first, the improved YOLOv5 model has excellent detection speed and detection accuracy and can enhance the real-time detection and positioning of cattle. Second, the improved YOLOv5 model is very lightweight, reducing the dependence on hardware configuration and computing costs. After verification in the real monitoring scene, it is proven that the fully autonomous intelligent UAV can help to reliably recover the single cattle identification from the air, through the standard deep learning pipeline and with the help of biometrics. The autonomous recognition method based on airborne depth reasoning proposed in this paper is very important for the population evaluation of large herbivores (such as Tibetan wild ass, Tibetan gazelle, rock sheep, etc.) in the source area of the three rivers. This non-contact real-time monitoring method is worth being popularized for the effective protection of local endangered species and the healthy development of the ecological environment.

**Author Contributions:** W.L. took charge of the conceptualization and writing, preparing original draft; Z.Z. (Ze Zhang) was responsible for methodology; X.L. (Xuqing Li) took charge of the software; H.W. took charge of the validation. The formal analysis was conducted by D.W. and Y.L., and the investigation was performed by K.L. and X.L. (Xueli Liu). S.Z. took charge of data curation. Y.Z. was responsible for writing—review and editing. Visualization was carried out by G.W. and Z.Z. (Zihui Zhao). The supervision was conducted by Y.H., P.F. and Q.S. took charge of funding acquisition. The published version of the manuscript has been read by all authors and their agreement was obtained. All authors have read and agreed to the published version of the manuscript.

**Funding:** This research was funded by the National Natural Science Foundation of China (no. 42071289); Innovation Fund of Production, Study and Research in Chinese Universities (2021ZYA08001); National Basic Research Program of China (grant number 2019YFE0126600); Major Special Project: The China High-Resolution Earth Observation System (30-Y30F06-9003-20/22); and Doctoral Research Startup Fund Project (BKY-2021-32; BKY-2021-35).

**Data Availability Statement:** Data for this research can be found at the following data link (https://pan.baidu.com/s/1mIdkhHGecOsP_d_fSeGl5w?pwd=xq5m. doi:10.11922/sciencedb.01121).

**Acknowledgments:** This research was completed with the support of Chengdu Bobei Technology Co., Ltd., Chengdu, China.

**Conflicts of Interest:** The authors declare no conflict of interest.

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
