# Peer review of "Intelligent Grazing UAV Based on Airborne Depth Reasoning"

_remotesensing, doi:10.3390/rs14174188_

Round 1
Reviewer 1 Report
Using intelligent UAV to identify and locate livestock through non-contact real-time monitoring is a new idea. This paper presents an improved algorithm of Yolo model combined with airborne computing unit, which uses LRCN visual framework to distinguish cattle in video. The route is reasonable and the work is solid. Here are some suggestions for further improvement:
1. There are obvious errors in Table 2: the AP and FPS columns are reversed, and the value and unit of the size of model column are wrong, so it is recommended to verify;
2. There is no discussion on the size of model in Table 2, and it is suggested to supplement it;
3. Please explain why CBAM will lose information when using MLP structure to extract channel information.
4. There are some typos in the paper. The authors should be check again, such as “and has become a remote sensing” in line 68, “including the spot patterns in manta rays [Error! Reference source not found.],” in line 84, “[Error! Reference source not found.].” in lines 94, 96, 141, “299 × 299 images” images? in line 99,
5. Figure 4 is not clear.
6. In “4.2.2. Recognition results”, does the results come from “NVIDIA Jetson AGX Xavier embedded platform”? I think the author should discuss the reasoning time in embedded devices.
Reviewer 2 Report
Thank you for the chance to review this interesting manuscript. The study concerns the modification and training of object detection algorithms to improve upon protocols for recognition of livestock from UAV imagery in real time. The paper employs innovative methods and demonstrates impressive accuracy of animal classification and animal recognition. However, as it currently stands the paper needs to improve in terms of clarity to ensure a reader can easily understand the study and results, and, furthermore, english language/grammatical errors need addressing throughout.
General.
1. The methodology of the paper uses a large number of abbreviations and technical language. Whilst partly unavoidable, I would encourage the authors to try to give explanations in more readily understood language wherever possible to ensure readers can follow the methods whether they are expert in this particular subject or not.
2. Please be consistent on use of capitals in 'yolov5' or 'YOLOv5' - I believe YOLOv5 is more correct since the YOLO part is an abbreviation.
3. Throughout, the word 'the' is heavily overused. For instance L40 could simply say '...have proved that sensor and information technology development assists...' (i.e. no 'the' required before 'sensor'). Please check and revise the entire manuscript for this to improve readability.
4. A key part of your title is the term 'airborne depth reasoning' yet this term is not defined and barely referenced within the paper. If it is to be in the title please explain the meaning of this term early on. Furthermore please consider if this is an accepted term. I can find only a handful of other papers using this term and one is another paper by the same lead author. Finally, later in the paper it is referred to as 'airborne deep reasoning' - please be consistent (deep or depth) if the term is to be used.
Specific.
L22-24. Please rephrase your opening statement - state more explicitly what is lacking from existing precision grazing technology. Using the term 'collectivization' (meaning: to apply to a group of people as a whole rather than individuals) requires further explanation e.g. 'existing livestock recognition technologies are not capable of identifying individual animals in real time'.
L27-28. Please reword and also expand on your methods statement in the abstract, it does not read as a complete sentence and is also not sufficiently descriptive of the techniques employed in your paper. A slightly longer overview would help the reader understand your study from the start.
L29. It is standard practice to write in the past tense in scientific documents. Please replace 'is' with 'was' and review whole document for similar errors.
L30-32. Suggest saying 'dynamic livestock targets for agricultural and ecological purposes'. When stating the conclusions in the abstract please keep them fairly specific to your findings and not too general.
L37. Suggest using 'grazing ruminants' rather than 'grassland ruminants'
L43. ‘legal’ not ‘legally’
L47-48 'Besides, people have concerned the animal welfare in terms of the ear-tagging' - please correct grammar in this sentence
Abstract: please reduce use of acronyms in the abstract as a casual reader would not be familiar with your abbreviated terms, please spell them out if used in the abstract e.g. KCF/LRCN etc.
L67-69. I do not think listing the journals in which such tools have been reported is very useful. Perhaps find a different way of saying this e.g. "Manuscripts in a range of prestigious journals have confirmed that deep learning is an important tool in the field of big data processing [citations]".
General. Please note that I am seeing some errors in the citations on L84,94,96,141 etc where the citation reads 'error! reference source not found'. Please fix.
L101. Please spell out KCF at first use
L94 - 120. These paragraphs give too much detail on your methodology to be in an introduction. They are more suited to methods. Please conclude your introduction with a simple statement on the aims/goals of your study which can include a brief mention of the algorithms to be used and why. Please also be clear on which aspects of your research are novel and which have been done before.
L109 - 120. It is not relevant to divide the work into chapters as the manuscript you present here is not divided in such a fashion. This reads like an excerpt from a thesis.
Fig1. It is not clear where exactly in figure 1 the study took place. Please mark it using either a point or polygon. Also I assume the coloured townships make up Qilian County however this is also not immediately obvious - perhaps it could be marked via an arrow/a thicker border applied around its edges/stated in the figure caption?
L128. remove "between"
L137. please mention how you selected the area to be flown by UAV within Qilian county. How large was the surveyed area and did you have some criteria used to select a survey area that was representative of the region?
L148. ROI was already defined in the introduction, please double check throughout that you define all abbreviations at first mention (and not define them a second time).
L138 instead of 'a large number', please state exactly how many and please also confirm some more details about the flights e.g. did they all take place on the same day, or over time and if so how much time? Also what was the flight height of the UAV, or did it vary? Did you conduct any radiometric correction of the flight images for varying light conditions?
L176. please spell out the wording of the abbreviation 'LSTM'
L192. Rather than saying 'amovlab' which is the company name, please state what is the piece of equipment you are referring to - this sentence is not clear.
L194. please define ROS
L205-208. please indicate the flight time that the laden-P600 can achieve with one set of batteries so the reader can determine the usefulness of the technology to monitor animals in extensive systems where large land areas may need to be covered to locate the animals.
L239. do you mean to say the individual cattle ID is required? how does this technique fare when applied to new herds, must new training data to identify the individuals be gathered before such a technique would be usable? This is something that should be remarked on in the discussion.
Fig 4. I note you include a LIDAR attachment amongst the hardware. Was the LIDAR attached during the flights conducted for this study? If not then you may add [optional] or similar tag next to the LIDAR to indicate a component that is not relevant to the present work.
L224. How does the pod control command manage when there are multiple moving objects within frame as might be the case with cattle in a herd? Is it still able to track individuals effectively or should the UAV be positioned so only a single animal is in frame?
L235. please state the name and manufacturer of the professional labelling software utilised
L246-247 If I am understanding correctly, these additional datasets add an additional 25,000 images to the original training dataset of 7000 images (ie actually making up about 80% of the training dataset), I think more description of the studies in which these images were collected (e.g. location, animals, equipment) should be included - possibly right from the start in section 2.1
Fig 5. Figure 5 is not readily understandable to anyone not intimately familiar with YOLOv5 (many abbreviations, explanations are available for some in the text but not all). Furthermore it is not clear which parts are the improvements and which are part of the original framework (could be indicated with different colour boxes or similar?). please try as best you can to make this figure understandable to any reader. Please also consider the same with regards figures 6 and 7.
L248. You mention several datasets which are combined and you also mention additional data expansion - once these processes are complete exactly how many images (and individual cattle instances) were there in the final training dataset?
Fig6. part a of figure 6 has a spelling error in 'module'
L346-357. please format (italics, subscripts) consistently for letters that indicate variables e.g. n and fi
L367. I suggest that the information in 4.1 should be in the methodology section rather than the results section.
Results. It is not clear enough in your results when you are assessing classification and when you are assessing recognition. Suggest using the terms classification (e.g. a cow being correctly classed as a cow) and recognition (for an individual animal being recognised by its ID) very consistently and explicitly throughout the paper to prevent confusion between the two processes.
L388. please define IOU
L390-391. please check grammar of the sentence on FNs, it is not clear.
L412-414. please check your spelling of YOLOv5 here and throughout.
415. section 4.2.2. is only one sentence long. I suggest this could be combined into another section, or expended upon.
L399. please also define how F1 score is calculated as you have done with your other metrics.
L425. I do not think the size of the differences reported in table 2 are enough to justify saying that the improved model has 'significant' advantages. My conclusion from table 2 would be that the modified YOLOv5 offers a slight advantage over the original YOLOv5.
Table 2. the values in the table and in the text (L425) for AP and FPS do not agree - one or the other is switched around.
Table 2. what is the 'size of model' in table 2 and how is it calculated.
L494. suggest saying '...positioning of cattle using UAVs...'. It is the cattle that are detected, not the UAVs themselves!
L496. is the UAV truly fully autonomous? What about take-off/landing and movement to the area the cattle are in?
Round 2
Reviewer 2 Report
The authors have taken on board my original comments and made a number of improvements to the text that have improved the manuscript’s clarity and readability - particularly in the abstract and in the methods. However, I still have some minor concerns that I would like to see addressed.
General:
1) Although the authors have improved the clarity of the document in this iteration, I still feel that English grammar in the paper is not yet at a standard suitable for publication, particularly in the introduction and the methodology. I suggest that the authors should have some proofreading done before submitting a final version.
Specific
L95. Please define ML at first use
L95. extraction is a single word - please remove the space
L151. I suggest finishing your introduction with one or two sentences to quickly summarise the goal of the present study. E.g. 'Therefore, he goal of the present work was to demonstrate...'
Fig 1. Thanks for updating figure 1 with some additional info. However, the figure part 1d contains a watermark, is this an image that was obtained from your UAV or from some other source as an example? Please ensure you are using images without watermarks. Furthermore, parts 1b and 1c of the figure show the following legend items 'GPS' and 'subplot boundary'. Please explain these items either in the figure caption or in the main text.
L234. please use capitals for 'ID"
Fig 6. thanks for including some colour in Fig 6. Please add to the figure caption to clarify the meaning of the colour e.g.' coloured boxes indicate areas where modifications were made'. Do the different colours mean different things?
L397. I think you mean to reference figure 6 here (as figure numbers have been updated).
Fig 10. Please make the legend match the capitalisation in the rest of the text.
L540-541. Please check the numbers given on these lines for FPS and AP of the modified YOLOv5 as they are still not matching table 2. for example the table states AP for the modified YOLOv5 is 0.80 and the text states 98.3%?
Table 2. Since you are using abbreviations in a table, normally these would be defined as a foot note to the table even though you have also defined them in the main text. This helps tables to be understood without reference to the main text.
L548. use past tense, 'it generated', not 'will generate'.
L581-582. Further to my previous comment about Table 2 not matching the text. This is another instance in which the numbers given are not matching the table. Please check. Also when giving a comparison like this one please state what is compared against what is it modified YOLOv5 versus original YOLOv5 or versus the faster RCNN?
L611. please use 'verification' not 'verified'
Author Response
Please see the attachment
This manuscript is a resubmission of an earlier submission. The following is a list of the peer review reports and author responses from that submission.